# Benefits from Repetitive Transcranial Magnetic Stimulation in Post-Stroke Rehabilitation

**DOI:** 10.3390/jcm11082149

**Published:** 2022-04-12

**Authors:** Michał Starosta, Natalia Cichoń, Joanna Saluk-Bijak, Elżbieta Miller

**Affiliations:** 1Department of Neurological Rehabilitation, Medical University of Lodz, Milionowa 14, 93-113 Lodz, Poland; elzbieta.dorota.miller@umed.lodz.pl; 2Biohazard Prevention Centre, Faculty of Biology and Environmental Protection, University of Lodz, Pomorska 141/143, 90-236 Lodz, Poland; natalia.cichon@biol.uni.lodz.pl; 3Department of General Biochemistry, Faculty of Biology and Environmental Protection, University of Lodz, Pomorska 141/143, 90-236 Lodz, Poland; joanna.saluk@biol.uni.lodz.pl

**Keywords:** post-stroke, rehabilitation, repetitive transcranial magnetic stimulation, noninvasive brain stimulation, motor function, cognitive function, depression, aphasia

## Abstract

Stroke is an acute neurovascular central nervous system (CNS) injury and one of the main causes of long-term disability and mortality. Post-stroke rehabilitation as part of recovery is focused on relearning lost skills and regaining independence as much as possible. Many novel strategies in neurorehabilitation have been introduced. This review focuses on current evidence of the effectiveness of repetitive transcranial magnetic stimulation (rTMS), a noninvasive brain stimulation (NIBS), in post-stroke rehabilitation. Moreover, we present the effects of specific interventions, such as low-frequency or high-frequency rTMS therapy, on motor function, cognitive function, depression, and aphasia in post-stroke patients. Collected data suggest that high-frequency stimulation (5 Hz and beyond) produces an increase in cortical excitability, whereas low-frequency stimulation (≤1 Hz) decreases cortical excitability. Accumulated data suggest that rTMS is safe and can be used to modulate cortical excitability, which may improve overall performance. Side effects such as tingling sensation on the skin of the skull or headache are possible. Serious side effects such as epileptic seizures can be avoided by adhering to international safety guidelines. We reviewed clinical studies that present promising results in general recovery and stimulating neuroplasticity. This article is an overview of the current rTMS state of knowledge related to benefits in stroke, as well as its cellular and molecular mechanisms. In the stroke rehabilitation literature, there is a key methodological problem of creating double-blinding studies, which are very often impossible to conduct.

## 1. Introduction

Stroke is an acute neurovascular central nervous system (CNS) injury and one of the main causes of long-term disability and mortality [1]. Brain damage leads to an imbalance in blood supply and may lead to dysfunctions affecting movement, sensation, cognition, speech, emotion, etc. Since 1947, when Physical Medicine and Rehabilitation became a board-certified specialty, there has been a big interest in introducing new therapies that not only can help patients in treatment-specific organs or systems, but also improve global functioning. This unique approach to patient expectations allows the creation of a special model of personalized treatment with a multi-specialist rehabilitation team [2]. Post-stroke rehabilitation as a part of recovery is focused on relearning lost skills and becoming as independent as possible. Recently, many novel strategies in neurorehabilitation have been introduced. One of them is transcranial magnetic stimulation (TMS), which, as a noninvasive brain stimulation (NIBS), is used in neurological and psychiatric conditions, especially in Alzheimer’s disease (AD), mild cognitive impairment (MCI), depression, and mental disease [3,4,5,6,7].

Different types of NIBS have been introduced over the last years, including electroconvulsive therapy (ECT), transcranial alternating current stimulation (tACS), magnetic seizure therapy (MST), TMS, and transcranial direct current stimulation (tDCS) [8,9].

However, TMS and tDCS are the most common modalities [10]. TMS is a more specific tool than tDCS because of its direct impact on brain networks. Moreover, its clinical effectiveness was proven in the treatment of mental disorders. Among the different types of NIBS, TMS, and especially repetitive TMS (rTMS), is a highly precise therapy with good control of stimulation parameters (frequency and location) [8]. TMS’s ability to inhibit neurons by low-intensity and activate neurons by high-intensity is very useful. Neurobiological mechanisms of TMS action are not fully understood; however, several hypotheses have been suggested [8,11]. For example, Cambiaghi et al. [12] investigated the effects of high-frequency (HF) rTMS treatment on morphological plasticity of pyramidal neurons in layer II/III (L2/3) of the primary motor cortex in mice.

Accumulating data shows that decreased corticospinal excitability in the affected hemisphere enhances intracortical excitability in the whole brain and modulates interhemispheric interactions. Generally, stroke recovery is based on two opposite models of neuroplastic reorganization: interhemispheric competition and vicariation. The first model, interhemispheric inhibition, presents the balance between brain hemispheres as a healthy condition, with the huge imbalance during injury (stroke) causing overexcitability of the unaffected hemisphere. Thus, reducing the activity of the unaffected hemisphere is necessary to match the higher inhibition of the affected hemisphere (ipsi- and contralateral brain inhibition). The second model, called vicariation, is hypothesized as an activity in the unaffected hemisphere that is focused on a compensation mechanism, taking over work from the damaged area; therefore, low-frequency stimulation for the affected hemisphere is needed. These two models of reorganization lead to opposite conclusions about optimal neuromodulatory strategies. There is a question of which method is better for a particular patient, inhibitory or excitatory. Describing structural reserve, Pino et al., proposed to combine these two models into one by a new mode-bimodal-balance model of recovery [13]. The advantage of TMS therapy is a painless, noninvasive method with the ability to modulate cortical excitability in the stimulated area as well as in remote points, promoting functionally related regions of the brain. Generally, collected data suggest that HF-rTMS (5 Hz and beyond) produces an increase in cortical excitability (size of induced MEPs during rTMS stimulation), whereas low-frequency rTMS (≤1 Hz) decreases cortical excitability [14].

In this review, we focus on the clinical benefits of TMS in post-stroke rehabilitation, with better results in aphasia, cognition, motor improvement, dysphagia, etc. The literature search was done using MEDLINE, EMBASE, and PEDro databases. In total, fifty-four articles were analyzed, including 33 randomized controlled trials, original research papers, and reviews (meta-analyses, systematic reviews, literature reviews). We included 93 articles since 2005, of which 46 were published in or after 2017. All the articles evaluated different benefits from using rTMS in stroke patients and were published in English. Search terms included: repetitive transcranial magnetic stimulation, noninvasive brain stimulation, rehabilitation, post-stroke, motor function, cognitive function, depression, aphasia. Moreover, we discuss the neurobiological basis (cellular and molecular mechanisms) of TMS in the brain and factors that can have an impact on post-stroke recovery.

## 2. Neurobiology of TMS in Stroke

The exact mechanism by which TMS works on neurons is under constant investigation, but current knowledge suggests that changes occur at the level of individual neurons as well as throughout the neural network. In addition, effects occur immediately after stimulation as well as after a delay. The strength of an impulse used in therapy depends on the individual’s excitability threshold. On the other hand, the location of probe application is determined by simple anthropometric methods or by using modern neuronavigation techniques based on an image of the brain obtained by magnetic resonance imaging [15].

Selecting the stimulation intensity is an important step in determining the rTMS dose. There are two main intensity selection approaches. The first one is based on determining the stimulation intensity, such that the application of rTMS leads to the desired effect. 

The aim of the second is to provide reproducibility, where standardization of parameters is crucial across experiments and information must be reported for a given protocol. The overwhelming majority of studies do not report sufficient information to reproduce the stimulation intensity and, eventually, the dose of rTMS. There is a need for developing a standardized and easy-to-follow reporting guideline to document the most crucial stimulation parameters in the best possible way [16]. Depending on the intensity and frequency of stimulation, the number of pulses and the pause time between them, the targeted area of the brain, and the type of coil, a different biological effect is achieved by TMS. The first mechanistic study of the effects of TMS in humans was carried out by Tofts [17], who suggested a model for the distribution of currents in the CNS, which assumed that neurons located horizontally in a surface parallel to the coil and the brain are preferentially stimulated by TMS. It has been observed that the rapid change of the magnetic field induces circular electric currents, thus the current flow is parallel to the coil and to the scalp on which the coil is placed flat. The strength of the magnetic field produced by TMS, despite being diminished by extra-cerebral tissues such as bone, scalp, and meninges, produces a magnetic field, leading to axonal depolarization and activation of cortical pathways. Importantly, some subcortical structures, among others the thalamus and basal ganglia, are not stimulated by TMS because of the lower impedance of white matter compared to that of gray matter [11]. Activation of the motor cortex by TMS causes the generation of descending volleys in the pyramidal system of salient corticospinal tracts created by spinal motor neurons. Corticospinal streams, stimulating motor neurons, lead to motor-evoked potential (MEP) registered by an EMG. The recruitment of motor units takes place in an orderly manner according to the principle of size, from the smallest to the largest [18]. Descending corticospinal activity, as manifested by a series of descending cortico-spinal salvo, is triggered by a single TMS pulse. The earliest wave (D-wave) has been found to be caused by direct stimulation of the rapidly conducting neurons of the pyramidal pathway. On the other hand, indirect, transsynaptic stimulation of the pyramidal pathway is caused by the I-wave [19].

A circular coil is used to activate superficial and large motor regions, including the upper limbs, because the electric field generated by it spreads widely and stimulates the superficial cortical layers. On the other hand, the current generated by double-cone coils has the ability to reach deeper cortical layers, hence these coils are recommended for stimulation of the lower-limb movement areas, located in the inter-hemispheric fissure. Moreover, figure-of-eight coils enable a more precise determination of the place of action, since the maximum point of the generated field is reached below its center [20]. The direction of the electric current depends on the position of the coil over the sulci and gyri of the brain, as well as on the orientation of the coil. The parallel arrangement of the figure-of-eight coil to the inter-semicircular fissure above the area of the primary motor cortex (M1) generates a posterior–anterior current leading to indirect stimulation of the pyramidal pathway through the excitation of interneurons. On the other hand, the perpendicular arrangement of the figure-of-eight coil to the interhemispheric fissure generates mainly an early I-wave, being the later wave induced by indirect transsynaptic stimulation of the neurons of the pyramidal pathway, or a D-wave, which is the earliest wave induced by direct activation of this pathway [21].

### 2.1. Cellular Mechanism of rTMS Action

The most widespread mechanism of action of rTMS is the effect on functional changes in synapses, mainly long-term potentiation (LTP) and long-term depression (LTD) [22]. LTP is produced by high-frequency stimulation (5–20 Hz), while LTD is generated by low-frequency treatment (1 Hz). Both presynaptic and postsynaptic mechanisms are involved in the induction of LTP and LTD, leading to depolarization of neurons and activation of the N-methyl-D-aspartate (NMDA) receptor and voltage-dependent calcium channels, which causes the influx of calcium ions into the cell interior. The level of calcium ions regulates the activity of protein kinases, such as Ca^2+^/calmodulin-dependent protein kinase II (CaMKII), protein kinase A (PKA), protein kinase C (PKC), and phosphatases, which in turn control the function of receptors and other proteins [23]. The formation of plasticity involves postsynaptic changes in the phosphorylation of neurotransmitter receptors, synaptic transport of α-amino-3-hydroxy-5-methyl-4-isoxazolepropionic acid (AMPA) receptors, and clustering of receptors in the postsynaptic membrane. An extremely important role in the late phase of LTP (and also in some forms of LTD) is played by the synthesis of new proteins, which may partially take place near the synapses [24]. An important element in the LTP and LTD phenomena are changes in the shape and number of dendritic spines. In vitro studies using in vivo imaging of neurons during stimulation have shown that LTP is associated with spine enlargement, and LTD is associated with spine contraction [25]. It seems that under the influence of stimulation, mature neurons can also produce new dendritic spines, which first, in the form of dynamic filopodia, grow from the trunks of the dendrites and then form synapses on the already existing axonal endings, creating the so-called polysynaptic axonal terminals [26]. At the molecular level, structural plasticity is associated with dynamic changes in the organization of the actin cytoskeleton [24]. LTP enhancement of synaptic strength by TMS is limited in time, the effect being seen over several days/weeks/months, in contrast to the reduction of synaptic strength by LTD, which has been reported as long-term [27]. It has been shown that a rapid Ca^2+^ influx caused by activation of NMDA receptors in the postsynaptic membrane leads to LTP, while a slow Ca^2+^ influx induces LTD [28]. Several studies reported that low-frequency TMS (1 Hz) inhibited the generation of muscle response [29,30,31,32]. Pascual-Leone et al. [33] confirmed that the frequency and intensity of TMS determine the stimulatory or inhibitory effect of the motor cortex. Placing the TMS coil on the scalp above the thumb abductor stimulation site induced increasingly stronger MEP in other muscles (radial wrist extensor, bicep arm, and deltoid muscle). As the intensity and frequency of rTMS increased, the number of pulses responsible for this excitation spread decreased. The inhibition period of MEPs generated in other muscles was longer than that of a single TMS activation pulse, which may be the result of a delay in intracortical conduction along the cortico–cortical pathways. In addition, Chen et al. [29], in a study evaluating the effect of TMS on motor-cortex excitability in humans, found that 1 h TMS stimulation with a frequency of 0.1 Hz had no effect on motor cortical excitability. In contrast, in in vitro studies and animal models, 810 TMS pulses at a frequency of 0.9 Hz for 15 min reduced the amplitude of the MEP. LTD induced by TMS persisted for at least 15 min after the stimulation was stopped.

### 2.2. Molecular Mechanism of rTMS Action

The neurorestorative properties of TMS are related not only to the LTD/LTP-like effect but also to modulating the neuronal excitability of inhibitor neurons and the membrane potential. Furthermore, TMS stimulation regulates the excitability of white matter components, including astrocytes that modulate neuronal circuits and are sensitive to neuronal activity. Moreover, Ca^2+^ is involved in the interaction of neurons and astrocytes in the synapse [34,35].

Brain-derived neurotrophic factor (BDNF) is a key factor involved in neuroplasticity; it has an affinity for the TrkB receptor and activates the PI3K/Akt, PLCγ Ras/MAPK, and GTPase signaling pathways [36]. Its action is related to the effect on the neurite growth cones in the sympathetic ganglia as a chemoattractant. Together with chemorepellents, it determines the proper innervation of target sites [37]. Dependent on BDNF are, inter alia, ascending mechanoreceptor fibers, retinal ganglion cells, and hippocampal neurons. BDNF is involved in synaptogenesis between motor neurons and Ia afferent fibers within the spinal cord [17]. BDNF also increases serotonin turnover and influences the functions of the CNS serotonergic neurons. In addition, it increases the level of norepinephrine in tissues and can also act as a neuromodulator of synaptic transmission and affect glutaminergic and GABAergic neurons. In the presence of neurotrophin 4, BDNF affects cerebellar granule cells and midbrain dopaminergic neurons, as well as neurons of the vestibular ganglion [38]. Brain-derived neurotrophic factor plays an important role in the proper functioning of the hippocampus and neocortex [36]. Lowering its level through exposure to stress, depression, or due to advanced age leads to a deterioration of the ability to remember designated tasks. It has also been found that BDNF has a trophic effect on retinal ganglion cells and influences the formation of dendrites and axons by them [39]. Data from preclinical and clinical studies on the effect of rTMS on serum BDNF levels are contradictory [40,41,42,43]. Heath et al. [40] investigated the effect of rTMS of varying frequency on neurobiological markers and depressive behavior in a mouse model. Mice were stimulated with rTMS at a frequency of 20 Hz with varying intensity: 4 mT, 50 mT, and 1 T, for 3 min, 5 times a week for 4 weeks. It was shown that medium (50 mT) and high (1 T) rTMS intensity decreased psychomotor agitation and also caused changes in glutamate signaling and glutamine processing. In contrast, only moderate-intensity stimulation enhanced BDNF concentration and neurogenesis, and high-intensity rTMS normalized the plasma levels of 3-methylhistidine and α-amino-n-butyric acid. In turn, Chen et al. [41] determined the effect of combined rTMS (15 Hz, 1.26 T, for 7 day; 900 pulses per day) and quetiapine treatment on hippocampal proliferation and in vitro and in vivo expression of BDNF and signal-regulated phosphorylated extracellular protein kinase (pERK1/2), as well as depression-like behavior in Sprague–Dawley rats. They observed that rTMS and quetiapine enhanced the proliferation of hippocampal cells as well as the expression of BDNF and pERK1/2 in hippocampal-derived neural stem cells. Moreover, the intervention abolished depression-like behaviors in animals exposed to chronic stress. The long-term effect of rTMS (15 Hz for three weeks) on the mechanism of improving depressive behavior in a chronic, unpredicted, mild-stress rat model was assessed by Feng et al. [42] They showed that two weeks after the end of stimulation, the enhanced effect of hippocampal cell proliferation, increased BDNF concentration and ERK1/2 phosphorylation, as well as amelioration of depressive disorders by rTMS was preserved. Furthermore, Jiang and He [43] conducted a meta-analysis assessing the effectiveness of rTMS and the relationship with serum BDNF levels, which included 11 clinical trials involving a total of 278 people, including 100 healthy volunteers and 178 patients with CNS diseases. The rTMS stimulation differed in the parameters used. It was shown that without taking into account both rTMS parameters and demographic data of the participants, rTMS did not enhance BDNF concentration. It was observed that the age of the participants, medical history, rTMS frequency, and the duration of the intervention had the greatest influence on the effect of rTMS on BDNF. In turn, based on a subgroup analysis, it was shown that rTMS lowered BDNF levels in healthy participants, as did the use of high-frequency rTMS as opposed to low-frequency rTMS. Moreover, prolonging the intervention time resulted in an increase in BDNF concentration. The above data indicate that further research is required to establish a unified protocol for the use of rTMS.

The current state of knowledge suggests that apoptosis is the major type of neuronal death, particularly in the ischemic region, during stroke. In vivo studies in animal models of ischemia have demonstrated the anti-apoptotic and neuroprotective effects of rTMS [44,45,46,47]. Guo et al. [46] demonstrated that rTMS therapy reduced brain damage as well as ameliorated cognitive impairment in a rat model of ischemia by inhibiting apoptosis in the ipsilateral hippocampus as well as augmentation of neurogenesis. In this study, it was noted that in the ischemic hippocampus, the expression of both BDNF, TrkB, and the anti-apoptotic protein Bcl-2 was significantly increased under the influence of rTMS. On the other hand, the levels of the anti-apoptotic protein Bax and of TUNEL-positive cells decreased. Importantly, changes in the expression of BDNF and proteins involved in apoptosis were observed, both at the protein and mRNA levels. Similar data were obtained in a study by Yoon et al. [47], who investigated the neuroregenerative mechanism of rTMS (3500 pulses of 10 Hz for 2 weeks into the ipsileo cortex) in rats with subacute ischemic stroke. It was noted that in the study group, functionality improved, with a concomitant increase in Bcl-2 expression and a decrease in Bax expression. Moreover, no difference in NMDA and MAP-2 expression was observed between the groups, suggesting that the beneficial effects of rTMS are related to ischemic anti-apoptotic activity rather than neuroplasticity. It has also been suggested that the anti-apoptotic properties of rTMS are related to the inhibition of caspase 3 [48,49]. Caglayan et al. [48] demonstrated a beneficial effect of rTMS on ischemic damage by assessing the effects of rTMS (20 Hz) on perilesional tissue remodeling, locomotor activity, glial scar formation, axonal sprouting of corticobulbar tracts, and cell proliferation in a mouse subacute stroke model. It was found that, in the high-frequency rTMS-stimulated study group, infarct volume, inflammation, DNA fragmentation, and expression of genes involved in axonal sprouting were reduced, while cerebral blood flow and angiogenesis improved. Moreover, the beneficial effect of rTMS was associated with decreased activation of caspase 1, caspase 3, and Bax, with a concomitant increase in Bcl-xL expression. Importantly, the obtained results correlated with the formation of glial scars, neuronal degeneration, microglia activation, improved functional regeneration, neurogenesis and neuroplasticity of the pyramidal pathway, and tissue remodeling. On the other hand, in the study of the neuronal model of ischemia–reperfusion injury in vitro, the effect of rTMS frequency was investigated. High-frequency rTMS (10 Hz, 3 s rest for 10 min) has been shown to inhibit cell apoptosis, enhance proliferation by modulating the ERK and AKT pathways, and activate the CaMKII/cAMP-response element binding protein (CREB) pathway promoting synaptic plasticity and expression of trophic factors. There was no neurorestorative effect resulting from low-frequency rTMS (0.5 Hz, 3 s rest for 10 min) [50].

In recent years, non-neuronal cells have been of interest as a potential target of post-stroke neuroregeneration. Astrocytes have the ability to protect brain tissue and reduce the disability resulting from stroke or other ischemic diseases of the brain tissue [51]. Yang et al. [52] showed that the use of high frequency (20 Hz) rTMS inhibits the proliferation and activity of astrocytes by reducing the expression of glial fibrillary acidic proteins (GFAP) and neuronal nitric oxide synthase (nNOS), leading to pain relief in rats with chronic neuropathic pain. On the other hand, Zorzo et al. [53] found no effect of high-frequency rTMS on astrocytes and microglia density in the hippocampus of healthy rats. In contrast, rTMS has been found to support metabolic activity and c-Fos in the parietal and retrosplenial cortices. Recent reports by Hong et al. [54] indicate that rTMS (10 Hz) reduces astrocyte proliferation, stroke volume, and apoptosis, and reduces the level of TNF-α while increasing the level of IL-10. Thus, it has been suggested that rTMS enhances functional regeneration and neuronal plasticity after stroke by inhibiting the neurotoxic polarization of astroglia.

Some studies have proposed that TMS has beneficial effects on neurological deficits by modulating the neurotransmitter systems [55,56,57]. Clarkson et al. [55] showed that in the peri-infarction zone there is a tonic inhibition of neurons caused by the accumulation of γ-aminobutyric acid (GABA) as a result of impaired function of its transporter. Importantly, the use of an inverse agonist specific for the GABA(A) receptor at a delay after stroke significantly improved the functional state of mice with stroke. Data describing the effect of rTMS on GABA levels are sparse and inconsistent. Two small pilot studies in people with depression showed that GABA levels increased significantly after rTMS stimulation, but there was no correlation between GABA levels and depression scores [56,57]. Dubin et al. [57] observed that rTMS alleviated symptoms of depression, and in patients responding to treatment, the levels of GABA in the medial prefrontal cortex increased significantly. In contrast, Godfrey et al. [58] did not find an effect of rTMS on GABA levels in either the left dorsolateral prefrontal cortex or the right motor cortex in currently depressed individuals. It should be noted, however, that all studies were conducted on a small group of participants, and further studies are warranted to demonstrate the unequivocal effect of rTMS on GABAnergic transmission.

Based on preclinical and clinical studies, it has been suggested that stroke may cause changes in dopamine (DA) function. On the other hand, activation of its receptors may be a new potential therapeutic target for motor neurodegeneration in patients after stroke [59]. The synthesis and release of DA under the impact of rTMS has been extensively studied both in animal models and in humans [49,60,61,62,63]. Yang et al. [60] noted that rTMS (0.5 Hz) increased DA levels while promoting DA-dependent cell survival in a rat model of Parkinson’s disease (PD). Similarly, Kanno et al. [61] observed that rTMS (25 Hz) significantly enhanced DA production in the dorsolateral prefrontal cortex, implying an improvement in motor function in rat PD models. In turn, Ba et al. [49] showed that rTMS counteracts the reduction of the striatum DA and the tyrosine hydroxylase of dopaminergic neurons in the substantia nigra. Moreover, it has been demonstrated that rTMS reduced the levels of caspase 3 and pro-inflammatory factors in injured substantia nigra. Interestingly, it has been shown that prefrontal rTMS stimulation promotes the release of DA in the mesostriatal, mesolimbic, and striatal areas, showing potential antidepressant effects [62,63].

Serotonin plays an important role in modulating the hypothalamic–pituitary–adrenal system and also in influencing cognitive functions. Regulation of serotonin levels in the brain is related to serotonin transporters (SERTs), and selective serotonin reuptake inhibitors (SSRIs) may enhance post-stroke cognitive disorders [64]. A meta-analysis evaluating the effect of rTMS on anxiety disorders unresponsive to treatment with SSRIs and serotonin–norepinephrine reuptake inhibitors showed a strong effect of stimulation on the functional improvement of patients [65]. Baeken et al. [66] assessed the effect of high-frequency rTMS (10 sessions) in the left dorsolateral prefrontal cortex (DLPFC) on postsynaptic 5-HT(2A) serotonin receptor binding in patients with treatment-resistant depression. Compared to the group of healthy volunteers, the depressed group showed higher baseline rates of receptor binding in the left hippocampus and lower bilateral binding in DLPFC. Moreover, there was clinical improvement in the study group, as well as a positive correlation with the 5-HT(2A) receptor affinity indices in bilateral DLPFC and a negative correlation with the right hippocampal 5-HT 2A) uptake values. It is worth noting that rTMS is not the only method of noninvasive brain stimulation that has a direct effect on 5-HT firing. A recent in vitro study found that tDCS over the left dorsolateral prefrontal cortex, both anodic and cathodic, had an SSRI-like effect by inhibiting dorsal raphe nucleus 5-HT neurons [67].

## 3. Benefits of Transcranial Magnetic Stimulation (TMS) in Stroke

### 3.1. Motor Function

Deficits in motor function are mainly unilateral and are related to location and severity of the brain damage [68]. Neuroplasticity is the key process in motor-function relearning. The motor cortex is the target of the pulsed magnetic field applied by TMS, which improves brain metabolism and neural–synaptic activity [69]. Treatment with rTMS involves the repeated application of a single, transcranial, high-intensity magnetic impulse to precisely stimulate an appropriate brain area—depolarizing neurons and activating excitatory action potentials, which inhibits/excites cortical neurons. Excitatory effects are obtained by using high-frequency (>3 Hz) rTMS (HF-rTMS), and inhibition is by low-frequency (<3 Hz) rTMS (LF-rTMS) [70]. Repetitive transcranial magnetic stimulation (rTMS) is a potential tool for treatment of post-stroke motor dysfunction [71].

Varying degrees of limb dyskinesia are the most common consequence of stroke. Spasticity, usually defined as a velocity-dependent elevation of muscle tone because of amplified stretch reflexes, is manifested in 65% of stroke survivors. Spasticity limits the mobility of patients and may worsen long-term disability. Repetitive TMS (rTMS) is a train of TMS pulses that continuously acts on the local brain with constant stimulus intensity. Conventional rTMS patterns include low-frequency rTMS (LF-rTMS) and high-frequency rTMS (HF-rTMS). Contralesional LF-rTMS can inhibit and ipsilesional HF-rTMS can promote local cortical excitability [1]. Table 1 summarizes the possible benefits of TMS therapy on motor function in post-stroke patients. All original articles were estimated on a PEDro scale that includes 10 items corresponding to internal validity and interpretability in order to rate their methodological quality. The average methodological quality score of RCT reports is 5.4 points on the 0–10 PEDro scale. 

### 3.2. Cognitive Function

Post-stroke cognitive dysfunctions occurred in approximately 75% of patients, and half of them can achieve various degrees of cognitive recovery, while the others continue to suffer cognitive impairment or even deteriorate to vascular dementia. The persistent cognitive deficit will result in long-term consequences in the activities of daily living (ADL), such as community reintegration, quality of life (QOL), and even physical functions. Some of the most common cognitive impairments after stroke include problems with memory and poor judgment. Therapeutic strategies for cognitive dysfunctions vary from pharmacological to non-pharmacological, including computer programs for cognitive training, physical activity, and brain stimulation, such as tDCs and rTMS [3]. Indeed, rTMS is used more and more often in the clinical treatment of cognitive impairment, mainly to improve memory function and to treat hemispatial neglect syndrome [85].

Generally, most studies present a complex model of treatment, with the use of neurodevelopmental treatment, motor relearning program, and activities-of-daily-living training with rehabilitation or cognitive training concomitantly with rTMS [86,87,88,89,90]. Table 2 presents improvements in cognitive functioning after TMS in stroke patients.

### 3.3. Depression

Depression is one of the most important symptoms in stroke. It is estimated that about 40% of stroke survivors subsequently experience depression. Long-term depression can lead to poor recovery, prolonged hospital stay, reduced rehabilitation effectiveness, and increased mortality [3,95,96].

Currently, few studies assessed the hypothesis that rTMS has a therapeutic, antidepressive effect on the brain. This might be the effect of modulation in functional connectivity in the area of the frontostriatal network [97]. Cambiaghi et al. [98] indicated that 1 Hz rTMS modulates dentate gyrus morphological plasticity in mature and newly generated neurons. Furthermore, data provide some evidence of an association between the antidepressant-like activity of 1 Hz rTMS and structural plasticity in the hippocampus.

In patients with TRD, remote temporal hypoperfusion, which is related to functional connectivity between the DLPFC and the default mode network (DMN), was observed mainly in the medial temporal limbic areas [99]. Another hypothesis describes the relationship between depression subtype and the response of the prefrontal cortex [100,101]. Table 3 summarizes the positive impact of rTMS on the emotional status of stroke patients.

### 3.4. Aphasia

Aphasia occurs in 21–40% of people after stroke in the first month, with approximately half of these experiencing significant speech impairment within 18 months after stroke onset. The risk of aphasia after a first-life stroke is about 4%. Generally, stroke patients with aphasia have a higher risk of mortality and depression, greater difficulty in functional capacity, and lower quality of life. Nowadays, there is a lack of effective pharmacological or non-pharmacological therapies for aphasia. However, new rehabilitation strategies based on brain stimulation have been developed. One of the most promising ones is the TMS technique that can modulate the focal activity of the cerebral cortex and produce changes in cortical function. Collected data show modulatory effects of rTMS on cortical excitability as a potential therapeutic technique in poststroke aphasics. Several studies presented that stimulation with low frequency (1 Hz) rTMS in the anterior area of the right hemisphere, in Broca’s homologous area (pars triangularis), increased activity corresponding with language emission in patients with non-fluent postictal aphasia [105].

Generally, naming ability in patients with aphasia corresponds to the power of cortical activation in the left hemisphere, mainly the frontal region.

A crucial factor in the analysis of effects of inhibitory stimulation with rTMS on language processing is elapsed time; slow functional changes induced by inhibitory rTMS may occur over a period of months or years. The fMRI study demonstrated altered patterns at 3 months after stroke that continued up to 46 months post-stimulation [106]. Table 4 summarizes the positive impact of rTMS on the speech and swallowing in stroke patients.

## 4. Limitations and Contraindications to TMS

TMS is a technique that has been used in neurology for over 15 years for routine diagnosis of motor pathway conduction and in scientific neurological research. The mechanism of action of TMS is to send a high-intensity current through a coil, producing a short, vertically running transient magnetic pulse. Since the scalp, skull, and meninges offer little resistance to the magnetic pulse, it can penetrate the cortex and induce an electrical current there. Side effects such as a tingling on the skin of the skull or headache are possible. Serious side effects such as epileptic seizures can be avoided by adhering to international safety guidelines [115].

Considering current scientific research, participants reported mild headaches, mild or moderate neck pain, mild sleep disturbances, and mild mood changes (which were noted as improvements in mood) [95].

Low-frequency rTMS reduces the excitability of the healthy brain hemisphere, reduces inhibition of the affected brain hemisphere, and reconstructs a new competitive inhibition balance pattern between the left and right brain hemispheres, which is beneficial to the recovery of stroke patients, and the low frequency is safer and almost never causes epilepsy [86].

Case series and feasibility studies seem to indicate a therapeutic effect; however, randomized, sham-controlled, proof-of-principle studies relating clinical effects to activation patterns are missing [113].

## 5. Conclusions

Post-stroke rehabilitation as part of patient recovery is focused on relearning the lost skills and regaining independence as much as possible. Many novel strategies in neurorehabilitation have been introduced. One of these is transcranial magnetic stimulation (TMS), which is used as a noninvasive brain stimulation (NIBS) in neurological and psychiatric conditions. This review provided a comprehensive overview of the cellular and molecular mechanisms of rTMS action and general benefits in stroke rehabilitation that may facilitate future decisions in patient therapy and trigger future innovative clinical studies. Commonly used in clinical practice is a combination of functional and cognitive therapy. Therapy based on traditional neurophysiological methods is used, but noninvasive brain stimulation, rTMS, is a promising tool that could be used in post stroke patients. The first stage after stroke consists of various self-repair processes, which could form a basis for a patient’s recovery. An interdisciplinary team and other specialists must be involved in a long-term rehabilitation program. The degree of functional and cognitive impairments affects the final treatment results. Considering this, other therapeutic methods require further rigorous clinical trials. Despite the limited amount of scientific evidence, our review shows that repetitive transcranial magnetic stimulation is mostly beneficial to patients. Moreover, the location of stroke lesions should be considered for future clinical trials. In the stroke rehabilitation literature, there is a key methodological problem of creating double-blinding studies, which are very often impossible to conduct. Therefore, in our review, we used the PEDro scale, which is a more comprehensive tool to estimate the methodological quality of the neurorehabilitation literature. Location of stroke lesions should be considered for future clinical trials.

## Figures and Tables

**Table 1 jcm-11-02149-t001:** Summary of potential benefits in motor functioning after using TMS therapy after stroke in RCTs with a valid measure of the methodological quality of clinical trials using the PEDro scale.

Study, Year	Stage of Stroke	Application Area, High/Low Frequency	Outcome, Measures	Main Findings
Chieffo et al. [71], 2021 PEDro: none	RCT, *n* = 12, Chronic	Primary Motor Cortex, HF-rTMS	FMA, Ashworth Scale, 10 m Walk Test, 6 min Walk Test	Spasticity significantly decreased only after the real rTMS. Bilateral HF-rTMS combined with cycling improved lower-limb motor function.
Kim et al. [72], 2020 PEDro: none	RCT, *n* = 77, Subacute	Primary Motor Cortex, LF-rTMS	BBT, FMA, FTT, Brunnstrom, Grip Strength	Real and sham rTMS did not differ significantly among patients within three months post-stroke. Location of stroke lesions should be considered for future clinical trials.
Khedr et al. [73], 2005 PEDro: none	RCT, *n* = 26, Acute	Primary Motor Cortex, rTMS	Scandinavian Stroke, Barthel Scale, NIHSS	Real rTMS improved patient scores more than sham.
Kirton et al. [74], 2008 PEDro: none	RCT, *n* = 10, Chronic (paediatric)	Primary Motor Cortex, LF- rTMS	MAUEF	rTMS improved hand function in pediatric score.
Liepert et al. [75], 2000 PEDro: none	RCT, *n* = 13, Chronic	Primary Motor Cortex, TMS mapping	MAL, ADL	After treatment, the muscle-output-area size in the affected hemisphere was significantly enlarged, corresponding to greatly improved motor performance of the paretic limb.
Fridman et al. [76], 2004 PEDro: none	RCT, *n* = 9, Chronic	Primary Motor Cortex, Dorsal Premotor Cortex (PMd), Ventral Premotor Cortex (PMv), TMS mapping	MRC	The dorsal premotor cortex of the affected hemisphere can reorganize to control basic parameters of movement usually assigned to M1 function.
Kim et al. [77], 2016 PEDro: none	RCT, *n* = 43, Subacute	Primary Motor Cortex, TMS mapping	FMA, Barthel Index	Potential advantages in predicting motor and ambulation recovery.
Choi et al. [78], 2018 PEDro: none	RCT, *n* = 24, Chronic	Primary Motor Cortex, HF-rTMS	NRS, MI-UL, MBC	HF-rTMS could be used as a beneficial therapeutic tool to manage hemiplegic shoulder pain.
Long et al. [14], 2018 PEDro: none	RCT, *n* = 62, Subacute	Primary Motor Cortex, LF-rTMS HF-rTMS	FMA, WMFT	Exhibited improvement in terms of the FMA score and the log WMFT time at the end of the treatment and 3 months later. Better improvement was found in the LF-HF rTMS group than in the LF-rTMS and sham groups. Combining HF-rTMS and LF-rTMS protocol in the present study was tolerable and more beneficial for motor improvement than the use of LF-rTMS alone.
Cha et al. [79], 2017 PEDro: 7/10	RCT, *n* = 30, Subacute	No data, rTMS	MEP, Peak Torque, 10 m Walk Test	Improvement of motor function recovery.
Kirton et al. [80], 2016 PEDro: 7/10	RCT, *n* = 45, Perinatal	Primary Motor Cortex, LF-rTMS	AHA, Melbourne Assessment, PedsQL, ABILHAND-Kids, Grip Strength, BBT	Assisting Hand Assessment gains at 6 months were additive and the largest with rTMS + CIMT. Quality-of-life scores improved. CIMT and rTMS increased the chances of improvement as part of complex rehabilitation.
Noh et al. [81], 2019 PEDro: 6/10	RCT, *n* = 22, Subacute	Primary Motor Cortex, LF-rTMS	Brunnstrom Stage, FMA, MFT, Grip Strength, MEP	Distal upper-extremity function, as measured by MFT and grip power, was improved.
Tosun et al. [82], 2017 PEDro: 6/10	RCT, *n* = 25, Acute/Subacute	Primary Motor Cortex, LF-rTMS	fMRI	LF-rTMS with or without NMES seemed to facilitate motor recovery in the paretic hand.
Abo et al. [83], 2014 PEDro: 6/10	RCT, *n* = 66, No data	No data, LF-rTMS	FMA, WMFT	FMA was significantly higher in both groups after the 15-day treatment compared with the baseline. Changes in Fugl–Meyer Assessment scores and Functional Ability Score of Wolf Motor Function Test were significantly higher in the NEURO group than in the constraint-induced movement therapy group.
Malcolm et al. [84], 2007 PEDro: 5/10	RCT, *n* = 19, Chronic	Primary Motor Cortex, HF-rTMS	WMFT, MAL, BBT	Participants demonstrated significant gains on the primary outcome measures: WMF and MAL, and on secondary outcome measures including BBT. Participants receiving rTMS failed to show differential improvement on either primary outcome measure. CIT can have a substantial effect, it provided no support for adjuvant use of rTMS.

**Abbreviations**: RCT: Randomized controlled trial; rTMS: Repetitive Transcranial Magnetic Stimulation; FMA: Fugl–Meyer Assessment; BBT: Box and Blocks Test; FTT: Finger Tapping Test; NIHSS: The National Institutes of Health Stroke Scale; MAUEF: The Melbourne Assessment of Upper Extremity function; MAL: Motor Activity LOG; ADL: Activities of Daily Living; MRC: Medical Research Council; MI-UL: Motricity Indices Upper Limb; MBC: Modified Brunnstrom Classification; NRS: Numeric Rating Scale; WMFT: Wolf Motor Function Test; MEP: Motor Evoked Potential Testing; AHA: Assisting Hand Assessment; MFT: Motor Function Test.

**Table 2 jcm-11-02149-t002:** Summary of potential benefits in cognitive functioning after using TMS therapy in stroke patients in RCTs with a valid measure of the methodological quality of clinical trials using the PEDro scale.

Study, Year	Stage of Stroke	Application Area, High/Low Frequency	Outcome, Measures	Main Findings
Liu et al. [91], 2020 PEDro: none	RCT, *n* = 62, No data	No data, HF-rTMS	FIM, MMSE, TMT-A, DST, DS	Improves the performance in the activities of daily living and attention function.
Fregni et al. [92], 2006 PEDro: none	RCT, *n* = 15, Chronic	Primary Motor Cortex, LF-rTMS	JTT, sRT, cRT, PTT, MMSE, Stroop Test	rTMS increased the magnitude and duration of motor effects. rTMS dose used in the study was not associated with cognitive adverse effects and/or epileptogenic activity. Improvement of the motor function performance in the affected hand. Significant correlation between motor function improvement and corticospinal excitability change in the affected hemisphere.
Askin et al. [93], 2017 PEDro: none	RCT, *n* = No data, Chronic	Primary Motor Cortex, LF-rTMS	Brunnstrom, FMA, BBT, FIM, Ashworth, MMSE	Improvements in all clinical outcome measures except for the Brunnstrom Recovery Stages. FIM cognitive scores and MMSE were significantly increased, and distal and hand Modified Ashworth Scale scores were significantly decreased.
Kulishova et al. [94], 2014 PEDro: none	RCT, *n* = 92, Subacute	No data, LF-TMS	MMSE	Significant regression of motor deficits. Reduction in anxiety and depression. Improvement in MMSE test (TMS being significantly more pronounced than that of low-frequency magnetic therapy).
Li et al. [86], 2021 PEDro: none	RCT, *n* = 70, Acute/Subacute	Contralateral dorsolateral prefrontal cortex (DLPFC), LF-rTMS	MMSE, MoCA, MBI	Cognitive domains—visuospatial function, memory, and attention were improved. More-significant improvement with rTMS.
Li et al. [3], 2020 PEDro: none	RCT, *n* = 30, Acute/Subacute	Contralateral dorsolateral prefrontal cortex (DLPFC), HF-rTMS	MMSE, MoCA	More-significant improvement with rTMS. Improvements in cognition domains.

**Abbreviations:** RCT: Randomized controlled trial; FMA: Fugl–Meyer Assessment; BBT: Box and Blocks Test; FIM: Functional Independence Measure; MMSE: Mini-Mental State Examination; TMT-A: Trail Making Test-A; DST: Digit Symbol Test; DS: Digital Span Test; JTT: Jebsen–Taylor Hand Function Test; sRT: Simple Reaction Time; cRT: Choice Reaction Time; PTT: Purdue Pegboard Test; MoCA: Montreal Cognitive Assessment Scale; MBI: Modified Barthel Index.

**Table 3 jcm-11-02149-t003:** Summary of potential benefits in depression after using rTMS in stroke patients in RCTs, reviews, and pilot studies (PSs) with a valid measure of the methodological quality of clinical trials using PEDro.

Study, Year	Stage of Stroke	Application Area, High/Low Frequency	Outcome, Measures	Main Findings
Duan et al. [8], 2018 PEDro: none	Review	Contralateral Dorsolateral Prefrontal Cortex (DLPFC), LF-rTMS, HF-rTMS	None	A total 1000 rTMS pulses (5–10 Hz at 80–100% of resting motor threshold over the left DLPFC and 1000 rTMS pulses (1 Hz at 80–100% of rMT) over the right DLPFC for 10 days was used to treat depression after stroke.
Frey et al. [102], 2020 PEDro: none	PS, *n* = 6, Acute/Subacute	Contralateral Dorsolateral Prefrontal Cortex (DLPFC), HF-rTMS	HAMD, Rankin Scale, FIM, NIHSS	HAMD significantly decreased and persisted at the 3-month follow-up. No statistically significant difference in FIM, mRS, or NIHSS.
Sharma et al. [103], 2020 PEDro: none	RCT, *n* = 96, Subacute	Primary Motor Cortex, LF-rTMS	Barthel Index, FMA, Hamilton Depression, Rankin Scale, NIHSS	Low-frequency 1 Hz rTMS on the PMC together with conventional physical therapy produced a significant change in mBI scores.
Sasaki et al. [104], 2017 PEDro: none	RCT, *n* = 13, Chronic	Medial Prefrontal Cortex (mPFC), HF-rTMS	Apathy Scale, QIDS	The degree of change in the QIDS score was greater in the rTMS group than that in the sham stimulation group.
Gu et al. [96], 2017 PEDro: none	RCT, *n* = 24, Chronic	Contralateral Dorsolateral Prefrontal Cortex (DLPFC), HF-rTMS	BDI, HAMD, MI-UE, MI-LE, MBC, FAC	BDI and HAMD significantly decreased. No significant changes in MI-UE, MI-LE, MBC, and FAC.
Hordacre et al. [95], 2021 PEDro: none	RCT, *n* = 11, Chronic	Contralateral Dorsolateral Prefrontal Cortex (DLPFC), HF-rTMS	BDI, SSEQ	Improvements in depression compared to sham after rTMS. Findings support the possible use of rTMS in depression after stroke. The mechanistic role of theta frequency functional connectivity appears worthy of further investigation.

**Abbreviations**: PS: Pilot study; FMA: Fugl–Meyer Assessment; BBT: Box and Blocks Test; FTT: Finger Tapping Test; FIM: Functional Independence Measure; NIHSS: The National Institutes of Health Stroke Scale; HAMD: Hamilton Depression Rating Scale; AS; Apathy Scale; QIDS: Quick Inventory of Depressive Symptomatology; BDI: Beck Depression Inventory; MI-UL: Motricity Indices Upper Limb; MI-LL: Motricity Indices Lower Limb; MBC: Modified Brunnstrom Classification; FAC: Functional Ambulatory Category; SSEQ: Stroke Self-Efficacy Questionnaire.

**Table 4 jcm-11-02149-t004:** Summary of potential improvements in speech and swallowing after using rTMS therapy in stroke patients in RCTs, meta-analyses (MA), and clinical trials (CT) with a valid measure of the methodological quality of clinical trials using PEDro.

Study, Year	Stage of Stroke	Application Area, Frequency	Outcome, Measures	Main Findings
Mottaghy et al. [107], 2006 PEDro: none	MA	Wernicke’s Area Dominant Motor Cortex, TMS vs. rTMS	simple picture naming task	Single-pulse TMS facilitates lexical processes due to a general pre-activation of language-related neuronal networks when delivered over Wernicke’s area. rTMS over Wernicke’s area also leads to brief facilitation of picture naming, possibly by shortening linguistic processing time.
Li et al. [108], 2020 PEDro:none	MA	LF-rTMS, HF-rTMS	SMD 95%CI	After rTMS with both low- and high-frequency, there was significant improvement in naming, while understanding and repetition did not change. Low-frequency rTMS has significant short-term importance in the subacute phase of a stroke.
Kakuda et al. [109], 2011 PEDro: none	CT, *n* = 4, Chronic aphasia	Inferior Frontal Gyrus (IFG), LF-rTMS	ST, SLTA, SLTA-ST, WAB	Combined LF-rTMS and intensive speech therapy (ST) is a safe and feasible therapeutic approach. Improved language function in post-stroke patients with motor-dominant aphasia.
Barwood et al. [110], 2013 PEDro: none	RCT, *n* = 12, Chronic non-fluent aphasia	Pre-central Gyrus Contralateral Hemisphere, Brodmann area 45 in Broca’s area, LF-rTMS	Behavioral language measures	Changes observed up to 12 months post-intervention when compared to the placebo control group in naming performance, expressive language, and auditory comprehension. LF-rTMS has potential clinical application for language rehabilitation in chronic aphasia.
Abo et al. [111], 2012 PEDro: none	CT, *n* = 24, Chronic non-fluent/fluent aphasia	Inferior Frontal Gyrus (IGF), Superior Temporal Gyrus (STG), LF-rTMS	ST	Significant improvement in listening comprehension, reading comprehension, and repetition in non-fluent aphasia patients. Significant improvement only in spontaneous speech in fluent aphasia patients.
Lopez-Romero et al. [105], 2019 PEDro: none	RCT, *n* = 82, Chronic non-fluent aphasia	Inferior Frontal Gyrus (IGF), LF-rTMS	MRS, BI, BT	rTMS applied to the inferior frontal gyrus is a safe therapeutic alternative in patients with non-fluent aphasia. Statistically significant differences in the Boston test of auditory compression, denomination, and praxis; also occurred on the 30th day in the naming domains and reading.
Hu et al. [112], 2018 PEDro: none	RCT, *n* = No data, Chronic non-fluent aphasia	Right Hemispheric Broca’s area, LF-rTMS, HF-rTMS	WAB	LF-rTMS group exhibited marked improvement over the HF-rTMS group in spontaneous speech, auditory comprehension, and aphasia quotients. LF-rTMS produced immediate benefits that persisted long-term, while HF-rTMS only produced long-term benefits.
Waldowski et al. [106], 2012 PEDro: none	RCT, *n* = 26, Early stroke aphasia	Right Inferior Frontal Gyrus (RIGF), LF-rTMS	CPNT, BDAE, ASRS	rTMS subgroup with a lesion including the anterior part of the language area showed greater improvement primarily in naming reaction time 15 weeks after completion of the treatment. Improvement in functional communication abilities. LF-rTMS over the inferior frontal gyrus area in combination with speech and language therapy seems to be beneficial for patients with frontal language area damage, mostly long after after finishing stimulation.
Thiel et al. [113], 2013 PEDro: none	RCT, *n* = 24, Early stroke aphasia	Right Inferior Frontal Gyrus (RIGF), LF-rTMS	AAT	Global Aachen Aphasia Test score was significantly higher in the rTMS group. Patients in the rTMS group activated proportionally more voxels in the left hemisphere after treatment than before (difference in activation volume index) compared with sham-treated patients.
Lim et al. [114], 2014 PEDro: 6/10	RCT, *n* = 47, Subacute stroke with dysphagia	Pharyngeal Motor Cortex (PMC), LF-rTMS	FDS, PTT, PAS, ASHA NOMS	FDS and PAS for liquid during the first 2 weeks in the rTMS and neuromuscular electrical stimulation (NMES) groups were significantly higher than those in the conventional dysphagia therapy (CDT) group, but no significant differences were found between the rTMS and NMES group. No significant difference in mean changes of FDS and PAS for semi-solid, PTT, and ASHA NOMS. Results indicated that both low-frequency rTMS and NMES could induce early recovery from dysphagia; therefore, both could be useful therapeutic options for dysphagia stroke patients.

**Abbreviations**: RCT: Randomized Controlled Trial; MA: Meta-Analysis; CT: Clinical Trial; SMD: Standardized Mean Difference; ST: Speech Test; SLTA: Standard Language Test of Aphasia; SLTA-ST: Supplementary Test of SLTA; WAB: The Japanese version of Western Aphasia Battery; MRS: Modified Rankin Scale; BI: Barthel Index; BT: Boston Test; CPNT: Computerized Picture Naming Test; BDAE: Boston Diagnostic Aphasia Test; ASRS: Aphasia Severity Rating Scale; AAT: Aachen Aphasia Test; FDS: Functional Dysphagia Scale; PTT: Pharyngeal Transit Time; PAS: Penetration–Aspiration Scale; ASHA NOMS: American Speech-Language Hearing Association National Outcomes Measurement System.

## Data Availability

Not applicable.

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
