# Peer review of "Benefits from Repetitive Transcranial Magnetic Stimulation in Post-Stroke Rehabilitation"

_jcm, 2022, doi:10.3390/jcm11082149_

Round 1

Reviewer 1 Report

The authors have adequately addressed previous concerns, thus significantly improving the quality of this manuscript. I do not have further comments.

Reviewer 2 Report

The ms is now strongly improved and I have no concerns for its publication.

This manuscript is a resubmission of an earlier submission. The following is a list of the peer review reports and author responses from that submission.

Round 1

Reviewer 1 Report

Here below some comments for Authors:

  • This sentence needs proper citation since ref [3] is devoted only to postroke patients

“One of them is transcranial magnetic stimulation (TMS), which, as non invasive brain stimulation (NIBS), is used in neurological and psychiatric conditions, especially in Alzheimer's disease (AD), mild cognitive impairment (MCI),..  [3].”

-              This sentence needs proper citation since ref [4] is devoted to TMS.

“Different types of NIBS have been introduced over the last years, including electro- 40

convulsive therapy (ECT), transcranial alternating current stimulation (tACS), magnetic 41

seizure therapy (MST), TMS and transcranial direct current stimulation (tDCS) [4]. 42”

-51 row why the font is bigger? “Accumulating data showed”..

- Generally, collected data suggest that high-frequency rTMS (5 Hz and beyond) results in high pic .. “ PIC stand for?

- 75 row. “We included 75 articles sinces 2005 but most of them is after 2017 publicated.” Be more concrete, please. Most of them? (how many)?

-row 87-90 – the intensity of TMS should be more clearly presented (the non-familiar reader can think the intensity is solely based on MRI imaging).

-row 159 “Pascual-Leone et al. Confirme” – should be properly cited and here is the error big letter. It should be. “Pascual-Leone et al. [ref missing] confimed…”

-row 166 sentence should be properly cited “In addition, Chen et al, in a study evaluating  the effect of TMS on the motor cortex excitability in humans, found that ..” It should be “In addition, Chen et al. [reference missing], in a study evaluating  the effect of TMS on the motor cortex excitability in humans, found that ..”

-row 197 non-proper citation, the reference number is missing “Heath et al investigated the effect of rTMS of varying frequency..”

-row 204 non-proper citation the reference number is missing “In turn, Chen et al…”

-row 213 non-proper citation the reference number is missing “..assessed by Feng et al.”

-rows 217-219 the reference number for Jianga and is missing

- row 237 “Guo et al. Demonstrated that..” should be properly cited and here is the error big letter

  • Further, there are many non-proper citation of reference numbers from row 244 (we did not correct here in revision further). This should be done by the authors later through the whole paper.
  •  

-222-231 rows these data should be written shorter to the general audience not going into the detail with the statistical results but rather understandable conclusion

-row 320-321 is the statistical info from the reported study crucial here?

-row 338-339, then row 347-348..somehow too many sentences on the same topic-  high versus low-frequency protocol. It is hard to read and follow

-TMF therapy? (row 350)

-Table 1, Table 2 the reference number should be positioned next to the author but not below PEDro

- Table 1, Table 2 is somehow missing the duration of rTMS application/location of stimulation in relation to “injury” site,  and the reader should realize if all reported studies are done for the rehabilitation purpose or ? At some instances, there is the term “TMS mapping”

Reviewer 2 Report

In the present review the Authors describe the importance of rTMS for stroke treatment. It is an interesting and clear paper, but there are some issues that need to be better described or clarified.

Thus, I have these observations:

Major:

- In the title, it should be better reported "repetitive transcranial magnetic stimulation" and not only "transcranial magnetic stimulation"; 

Minor:

- In the Abstract, please avoid reporting more than once "repetitive transcranial magnetic stimulation (rTMS)";

- Introduction: when reporting the wide use of NIBS techniques in neurological and psychiatric conditions, in particular "TMS and tDCS are the most common modalities", the Neurology paper by Peruzzott-Jametti et al., should be cited (2013);

- Introduction: when stating "Neurobiological mechanisms of TMS action are not fully understood, however several hypotheses have been suggested [4,5]", please report the recent (Beh. Brain Res. 2021) outcomes on the use of high-frequency rTMS in the mouse motor cortex;

- A reference is missing: "Some studies have proposed that TMS has beneficial effects on neurological deficits by modulating the neurotransmitter systems (x).". Please fix;

- When reporting the effects of rTMS on serotonin, it should be reported that also tDCS has been recently reported to have a direct effects on 5-HT firing in the mouse model;

- When reporting the effects on depression, plese cite a recent paper on the effects of low-frequency rTMS in the mouse model (Int J Env Res Pub Health 2020);

- There are some typos around (link in the bibliography, character size, spaces) that need to be fixed.

Reviewer 3 Report

This article could provide a comprehensive and interesting overview of the field of non-invasive brain stimulation (NIBS) and stroke. However, before further reviewing the content of the manuscript, substantial English editing is required. The manuscript is unfortunately not intelligible in its current version.

Reviewer 4 Report

The authors deal with a fascinating and timely topic, although not entirely new, i.e., the benefits from transcranial magnetic stimulation (TMS) in post-stroke rehabilitation. To this end, they focused on current evidence of the effectiveness of repetitive TMS (rTMS) as one of the non-invasive brain stimulation techniques. Therefore, they presented specific interventions, such as low-frequency or high-frequency rTMS therapy on motor function, cognitive function, depression, and aphasia in post-stroke patients. Overall, accumulated data suggest that rTMS is safe and can be used to modulate cortical excitability and synaptic plasticity, thus improving overall performance. Few comments to the authors, requiring some revision.

Abstract: it should contain more technical details and further findings, including the study limitations.

Introduction: the sentence “One of them is transcranial magnetic stimulation (TMS), which, as non-invasive brain stimulation (NIBS), is used in neurological and psychiatric conditions, especially in Alzheimer's disease (AD), mild cognitive impairment (MCI), depression, mental disease” needs additional citations (e.g., PMID: 34482205; PMID: 32340195; PMID: 30065211). Moreover, the rationale for a new review should be emphasized.

Benefits of transcranial magnetic stimulation (TMS) in stroke: please include reference to the statement “Neuroplasticty is the key process in motor function re-learning. Motor cortex is the target of pulsed magnetic field applied by TMS, that improve brain metabolism and neural – synaptic activity” (e.g., PMID: 34276553).

General: although the quality of language is acceptable, an editing by a native speaker would be useful.